# Analysis of Corn Yield Prediction Potential at Various Growth Phases Using a Process-Based Model and Deep Learning

**DOI:** 10.3390/plants12030446

**Published:** 2023-01-18

**Authors:** Yiting Ren, Qiangzi Li, Xin Du, Yuan Zhang, Hongyan Wang, Guanwei Shi, Mengfan Wei

**Affiliations:** 1Aerospace Information Research Institute, Chinese Academy of Sciences, Beijing 100094, China; 2College of Resources and Environment, University of Chinese Academy of Sciences, Beijing 100190, China

**Keywords:** corn yield prediction, growth phase, WOFOST, deep learning

## Abstract

Early and accurate prediction of grain yield is of great significance for ensuring food security and formulating food policy. The exploration of key growth phases and features is beneficial to improving the efficiency and accuracy of yield prediction. In this study, a hybrid approach using the WOFOST model and deep learning was developed to forecast corn yield, which analysed yield prediction potential at different growth phases and features. The World Food Studies (WOFOST) model was used to build a comprehensive simulated dataset by inputting meteorological, soil, crop and management data. Different feature combinations at various growth phases were designed to forecast yield using machine learning and deep learning methods. The results show that the key features of corn’s vegetative growth stage and reproductive growth stage were growth state features and water-related features, respectively. With the continuous advancement of the crop growth stage, the ability to predict yield continued to improve. Especially after entering the reproductive growth stage, corn kernels begin to form, and the yield prediction performance is significantly improved. The performance of the optimal yield prediction model in flowering (R^2^ = 0.53, RMSE = 554.84 kg/ha, MRE = 8.27%), in milk maturity (R^2^ = 0.89, RMSE = 268.76 kg/ha, MRE = 4.01%), and in maturity (R^2^ = 0.98, RMSE = 102.65 kg/ha, MRE = 1.53%) were given. Thus, our method improves the accuracy of yield prediction, and provides reliable analysis results for predicting yield at various growth phases, which is helpful for farmers and governments in agricultural decision making. This can also be applied to yield prediction for other crops, which is of great value to guide agricultural production.

## 1. Introduction

Crop yield is essential to support government agricultural decision making, assist agricultural management practices and optimize resource utilization [1]. Corn is a staple food for more than 4.5 billion people, and the demand is expected to double by 2050 [2,3]. Therefore, reliable corn yield prediction and estimation are becoming increasingly important for ensuring food security and maintaining sustainable agricultural development [4].

The formation of corn yield is a complex biological process, and crop growth characteristics and sensitivities to different environmental events vary with growth stage. Corn yield is affected by the growth state features and environmental factors of each growth stage. Thus, analysing the importance of each growth stage and feature is of great significance for corn yield prediction.

At present, commonly used methods for yield forecasting can be summarized into two categories: process-based crop growth models and empirical statistical models [5]. Crop growth models are process-based and dynamic simulation models [6,7], which can dynamically simulate crop growth and the formation process of yield with the support of agronomic mechanisms. Crop growth models have been widely used in crop growth assessment and yield prediction [8,9]. The input of the model requires meteorological data, crop genotypes, soil characteristics, and field management measures. These parameters require calibration, which is relatively difficult to obtain [6]. Studies have shown that crop growth models can provide daily data of yield factors in the whole growth process and satisfactory end-of-season yield estimates once the required input data and parameters are provided [10]. However, the entire growth process simulated by the crop growth model is simply divided into two stages, the vegetative growth stage and reproductive growth stage, with the flowering stage as the boundary. There is no way to analyse the importance or influence mechanism of factors in other important growth phases. Empirical statistical models relate crop yield to a number of selected features, and are usually simple, easy to understand and need fewer parameter settings, thereby making them widely used in crop yield prediction [7]. Most current empirical statistical models are based on linear regression models, such as multiple linear regression, which cannot capture the nonlinear relations between the dependent and independent variables [10]. Machine learning and deep learning methods have the advantages of learning nonlinear relationships between features and yield, showing better performance for yield forecasting [4,11]. Thus, machine learning and deep learning methods provide alternatives to traditional regression approaches and have become highly recommended to manage the complicated relationships between different variables and crop yield [12,13]. However, they rely on a large amount of remote sensing data and field survey data, facing a certain degree of sample scarcity. On the one hand, the quality of remote sensing images is poor or missing during critical crop growth periods due to cloud occlusion or changes in satellite orbits. On the other hand, it is difficult to obtain field survey data. Thus, the accuracy of yield prediction is limited. Considering that these two methods have their own advantages and disadvantages, combining their advantages to achieve high-precision yield forecasting at various growth phases needs further research.

The overall goal of this study was to explore the ability of corn yield prediction in different growth phases by combining the crop growth simulation model and deep learning methods. This approach enhances the agronomic mechanism of yield prediction, providing strong data support through a crop growth model and analysing the yield prediction ability of yield features at each growth phase through deep learning. This study focuses on the following two issues: (1) assessing the importance of crop yield-related features and (2) analysing the prediction potential of corn yield at each growth phase. Providing a theoretical basis for the selection of growth phases and features for the construction of corn yield forecast models under specific scenarios will help to improve the efficiency and accuracy of corn yield prediction.

## 2. Study Area and Data

### 2.1. Study Area

Shandong Province is located on the eastern coastal area of China (Figure 1) and downstream of the Yellow River within the range of 34°22.9′~38°24.0′ N and 114°47.5′~122°42.3′ E; it is one of the main producing areas of food crops and cash crops in China. It has a warm temperate continental monsoon climate with an annual average temperature of approximately 11~14 °C and an annual average precipitation of approximately 550~950 mm [14]. Summer corn is an important food crop in Shandong Province. The planting area, yield and total production of summer corn in this area rank first among summer corn planting areas in the country [15], which plays an important role in ensuring the food security of the region and even the whole country.

### 2.2. Data

#### 2.2.1. Meteorological Data

Meteorological data in this paper come from the National Meteorological Information Center (http://data.cma.cn, accessed on 18 December 2022). A total of 95 meteorological stations in the corn planting area were selected, and the time range was from 1995 to 2015. The dataset contains daily meteorological observation data, including daily average temperature (°C), minimum temperature (°C), maximum temperature (°C), average wind speed (m/sec), precipitation (mm), average water vapor pressure (kPa) and photoperiod (h).

#### 2.2.2. Soil Data

Soil data were obtained using the 1:1,000,000 Chinese soil map from the Nanjing Institute of Soil Science, Chinese Academy of Sciences, which mainly includes soil spatial distribution, soil physical properties, soil chemical properties and soil nutrient data.

## 3. Methods

Based on the crop growth model supported by agronomic mechanisms, this study dynamically simulated the continuous change in each feature and yield during the whole growth period of corn under various growth scenarios and constructed a sufficient dataset of summer corn yield and its related features. Using historical meteorological data and empirical phenological information to determine the accumulated temperature threshold, the whole growth period of corn in this dataset was refined into multiple growth phases. By analysing the correlation between each feature and yield, a feature set was selected, machine learning and deep learning methods were used to fully explore the relationship between features and yield under each growth phase, and the yield prediction potential of summer corn at different growth phases and their combinations was systematically analysed. The overall methodological workflow is shown in Figure 2.

### 3.1. Multiscenario Dynamic Simulation of the Corn Growth Process

The WOFOST (World Food Studies) model was developed by Wageningen Agriculture University and the Center for World Food Studies (CWFS) [16]. It is a process-based mechanistic model that can simulate crop growth from emergence to maturity with specific meteorological, crop, soil and management parameters [17]. The WOFOST model has been widely used in many countries and regions [1,18]. This study used the PCSE 5.5 (Python Crop Simulation Environment, PCSE) software package under Python to operate the WOFOST model. The growth process of corn in various scenarios was analysed, fully covering the possible growth conditions of summer corn in the study area, which provided strong data support for this study.

The input data for the WOFOST model include weather, crop, soil and management parameters [19]. Some previous studies [20,21,22,23] have already completed parameter sensitivity analysis and localization calibration of WOFOST for summer corn in Shandong province, and provided a valuable reference for the calibration of WOFOST in this study.

#### 3.1.1. Weather Data

The weather data required by the WOFOST model include DAY (date, d), IRRAD (irradiation, kJ/m2/day), VAP (vapour pressure, kpa), TMAX (maximum temperature, °C), TMIN (minimum temperature, °C), WIND (wind speed, m/sec), RAIN (precipitation, mm), and SNOWDEPTH (snow depth, cm). There are no IRRAD data in the meteorological observation dataset (Section 2.2.1), and they need to be calculated using the Angstrom equation according to the photoperiod [24]. Finally, the weather data were pre-processed to the WOFOST input format.

#### 3.1.2. Crop Parameters

As shown in Table 1, for the crop parameters with high sensitivity, such as cumulative temperature from emergence to anthesis (TSUM1), from anthesis to maturity (TSUM2), and specific leaf area (SLATB1), we set a fixed step size in a reasonable range to generate pseudo varieties, expand the scope of simulated scenarios, ensure the complexity of the simulation dataset, and further adjust the values of the parameters according to the results of the pseudo parameters. Other crop parameters use the default value or calibration values from the relevant literature mentioned above.

#### 3.1.3. Soil Parameters

The main soil parameters required by the WOFOST model include soil moisture content at wilting point (SMW), saturation (SM0), and field capacity (SMFCF). The values of these parameters mainly depend on soil texture and structure. The proportion of loam in Shandong Province is approximately 75%, which covers three types: sandy loam, light loam and medium loam [25,26]. Soil parameters in the study area were determined by soil data (Section 2.2.2) and previous studies [22,27]. The main soil parameters are presented in Table 2.

#### 3.1.4. Management Parameters

Summer corn in Shandong Province is generally sown in early June [28,29], so the sowing date was set as 6.01, 6.10, 6.19 (early, middle and late sowing), and two irrigation and rainfed conditions were simulated. In this study, irrigation adopted the WOFOST potential mode, and rain feeding adopted the WOFOST water-limited mode.

The above weather, crop, soil and management data were input into the WOFOST model to dynamically simulate the continuous change process of corn growth, and the whole-period daily data of DVS (development stage), LAI (leaf area index), TRA (transpiration rate), SM (soil moisture) and each organ biomass were obtained. LAI is an important state variable of the WOFOST model that is part of many dynamic growth processes. Finally, more than 390,000 growth scenarios were simulated, providing adequate data support for subsequent analysis.

### 3.2. Refinement of the Development Stage (DVS) in WOFOST

The accurate extraction of the growth phase is conducive to a reasonable analysis of the spatiotemporal and interannual changes in crops and the improvement of the yield prediction model [30]. The WOFOST model quantitatively characterizes the growth and development stages of crops through DVS. However, it only divides the entire growing seasons into vegetative growth (emergence to flowering) and reproductive growth (flowering to maturity), and the growth time span is too long. Considering the lack of other important growth stages of crops, it is necessary to refine the growth stage.

When the effective accumulated temperature reaches the accumulated temperature required to complete a certain developmental stage, the growth period ends and the next growth period is entered. The effective accumulated temperature is the accumulation of the daily average temperature above the crop base temperature [31]. It is directly related to the growth rate and growth stage of plants, and is an important indicator to measure the heat conditions in the process of crop growth and development. In some studies, the effective accumulated temperature has been used to divide the growth phase [5,32].

This study used the accumulated temperature data that affect the growth and development of crops to refine the DVS in the WOFOST model. Jointing and milking maturity are important growth stages in the formation of corn yield [33]. Thus, the whole growth period of summer corn was divided into four phases: emergence to jointing, jointing to flowering, flowering to milk maturity, and milk maturity to maturity.

#### 3.2.1. Determination of Effective Accumulated Temperature Threshold

Using the historical meteorological data from 1995 to 2015 and empirical phenology data, the calendar days of each growth stage of corn were converted into growing degree days. The timing of a specific developmental stage of a crop is usually expressed in calendar days. By converting calendar time to thermal time (growing degree days), the length and duration of a crop’s developmental stage can be adjusted for temperature conditions in different years. The five main phenological periods of summer corn in Shandong Province are the emergence stage in late June, jointing stage in mid-July, flowering stage in early August, milk maturity stage in late August, and maturity stage in mid-September [34,35,36,37,38]. The calendar days were converted into growth degree days by calculating the effective accumulated temperature (Formula (1)), and the average value was determined as the standard accumulated temperature threshold required for this growth stage of summer corn in the study area.
(1)$$T_{e}=\left\{\begin{array}{cc} 0, & \left(T\leq T_{base}\right)\\ T-T_{base}, & \left(T_{base}<T<T_{max}\right)\\ T_{max}-T_{base}, & \left(T\geq T_{max}\right) \end{array}\right.$$
where $T_{e}$ is the effective accumulated temperature (°C). T is the average daily temperature. $T_{base}$ is the lower bound of the developmental critical temperature, and in this study, it was set as 8 °C. $T_{max}$ is the upper limit temperature of corn development, and in this study, it was 29 °C [39].

#### 3.2.2. Calculation of DVS

The WOFOST model simulates crop growth based on the theory of accumulated temperature [40]. It is believed that temperature plays a leading role in crop growth and development, and day length factors are considered later to adapt to the growth characteristics of different photosensitive crops. The effects of temperature and day length on the DVS of crops can be calculated by Formulas (2) and (3):(2)$$DVS=F_{pr}\frac{\int T_{e}}{TSUM_{j}}$$
(3)$$F_{pr}=\frac{P-P_{c}}{P_{0}-P_{c}},\;0\leq F_{pr}\leq 1$$
where $T_{e}$ is the effective accumulated temperature (°C). $TSUM_{j}\left(j=1,2\right)$ represents the effective accumulated temperature required to complete a developmental stage. $F_{pr}$ is the day length reduction factor. P is the actual day length. $P_{c}$ is the critical day length. $P_{0}$ is the best day length. With the continuous evolution of crops, the influence of photoperiod on the growth of modern crops is greatly reduced compared to earlier crops, and the effect of photoperiod is usually no longer considered in the simulation [19].

### 3.3. Feature Selection

The crop yield is affected by the growth state and environmental conditions of different growth phases. Based on the above simulation dataset, correlation analysis of three different types of features (growth state, water-related and temperature-related) and yield was carried out, and various types of features with the highest correlation with yield were selected into the optimal feature set. According to the four growth phases described above, the mean value or cumulative value of each feature in each growth stage was calculated, which characterizes the overall growth status of each growth stage. A total of nine features of three types were selected to participate in the optimization: (1) features related to crop growth state: the average value, cumulative value, maximum value and growth rate of the leaf area index (LAI) at each growth stage; (2) water-related features: cumulative values of transpiration rate (TRA), soil moisture (SM) and precipitation (PPT) in each growth stage; (3) temperature-related features: cumulative values of average temperature (Tmean) and maximum temperature (Tmax) in each growth stage. Then, correlation analysis was used to analyse the contribution and sensitivity of these nine features to yield at different growth phases, and the feature with the highest correlation with yield among the three types was determined to participate in the following yield forecasting.

### 3.4. Yield Prediction Methods

#### 3.4.1. Random Forest (RF)

RF, first proposed by Breiman [41], fits a set of models that first trains a multitude of decision trees and then obtains predictions by averaging or voting through all individual trees. The algorithm has good tolerance to noise and outliers, good scalability and parallelism for high-dimensional datasets, and high prediction accuracy [42]. In this study, the input of the RF model was the selected yield-related features, which were first normalized by the min-max method before being fed into the model. The output of the RF model was the predicted yield. The simulated dataset was divided into training samples and testing samples at a ratio of 9:1. Three hyperparameters, including the number of decision trees (n_estimators), the maximum depth (max_depth), and the number of features (max_features), were tuned in this study. The optimal hyperparameters were determined by grid search via cross-validation. Additionally, feature importance was evaluated using the mean decrease accuracy method, which randomly added noise interference to out-of-bag data, and important features that can lead to a large drop in accuracy.

#### 3.4.2. The Gated Recurrent Unit (GRU)

GRU is a recurrent neural network, and was first proposed in 2014 by Cho et al. [43]. In this method, the flow of information is controlled through gates (reset gate, update gate), which not only better capture the dependencies in a time series, but also effectively solve the gradient explosion or gradient disappearance problem when capturing long-term dependencies using conventional recurrent neural networks (RNNs) [44,45]. According to the law of crop growth, the formation of yield is a continuous time series process. The GRU model can learn the dependencies of time series data and its performance in the field of yield prediction deserves further study.

This study used the TensorFlow (GPU version 2.6) environment to build the GRU model. The input of the model was the time series features of various growth phases, which were normalized by the min-max method, and the output of the GRU model was the predicted yield. The simulated dataset was divided into training samples and testing samples at a ratio of 9:1. The optimizer (Adam, Adaptive Moment Estimation) was used to determine the optimal value of GRU layers, units, epochs, batch size, dropout and other parameters to find the optimal model. The architecture of the GRU yield prediction model is shown in Figure 3.

### 3.5. Experimental Design

The performance of yield factors in each crop growth stage is related to the formation of the final yield and determines the importance of each growth stage for yield prediction. In this study, the simulated dataset was divided into training data and testing data according to a ratio of 9:1, and various analysis results were obtained from the validation data. According to the advancement of the crop growth process, we designed a single growth stage, multiple growth stage and different combinations of features to forecast yield using machine learning and deep learning methods. We fully excavated and systematically analysed the yield prediction potential of each growth stage of summer corn.

#### 3.5.1. Forecasting the Yield by a Single Growth Phase

We explored the yield prediction ability that can be reached in a single growth stage of summer corn and its key features. Since the information of a single growth stage is insufficient, the features of crop growth state, water and temperature were all used for yield prediction, and the importance of the features was evaluated. For each of the four growth phases, the RF model was used to forecast the yield, and feature importance analysis was carried out.

#### 3.5.2. Combination of Multiple Growth Phases

Three different types of combinations with two growth phases, three growth phases, and whole growth phases were designed, which gradually increased the number of growth phases in sequence, and we obtained a total of six combinations, as shown in Table 3. In each combination, features were gradually added, from single crop growth state features to the combination of crop growth state features, water and temperature features.

#### 3.5.3. Accuracy Evaluation

In this study, the performance of the yield prediction model was evaluated using three different indicators: coefficient of determination (R^2^), root-mean-square error (RMSE) and mean relative error (MRE). Better performance is associated with a higher R^2^ and lower RMSE and MRE. The formula is as follows:(4)$$R^{2}=1-\frac{\sum_{i}{\left(x_{i}-y_{i}\right)}^{2}}{\sum_{i}{\left(x_{i}-{\bar{x}}_{i}\right)}^{2}}$$
(5)$$RMSE=\sqrt{\frac{\sum_{i=1}^{n}{\left(x_{i}-y_{i}\right)}^{2}}{n}}$$
(6)$$MRE=\frac{\sum_{i=1}^{n}\left|x_{i}-y_{i}\right|}{n\times x_{i}}$$
where $x_{i}$ and $y_{i}$ represent the actual yield and the predicted yield, and ${\bar{x}}_{i}$ is the mean of the actual yield.

## 4. Results and Analysis

### 4.1. Reasonability of Simulation Results

The WOFOST model was used to construct a complete simulation dataset by inputting meteorological data, crop parameters, soil parameters and management measures data. This dataset contains daily data of the dynamic growth process of corn under 390,000 growth scenarios. The histogram of the simulated yield is shown in the Figure 4. The corn yield in the study area is fully simulated, including high-yield and low-yield situations. The average value of the simulated yield is 6709.39 kg/ha. In addition, we collected the measured yield data of some stations from some studies [22,46,47], and obtained statistical yield information from the official website of Shandong Provincial Bureau of Statistics. As shown in Figure 5, most of them can be within the range of the simulation results, reflecting the reasonability of the simulated dataset.

### 4.2. Results of DVS Refinement

DVS in the WOFOST model was refined using the effective accumulated temperature, and the results are shown in Table 4. From the table, we can determine the effective accumulated temperature required for each growth phase of summer corn and obtain the estimated DVS values corresponding to each growth phase. The estimated DVS of jointing, flowering, milk maturity and maturity were in the range of 0.45–0.47, 0.95–0.99, 1.47–1.54 and 1.93–1.99, respectively. Among them, the actual DVS was 1 at flowering, and 2 at maturity; their estimation errors were less than 5% and 3.5%, respectively, which reflects the rationality of refining the DVS using the accumulated temperature threshold.

In this study, five important phenological periods from the entire growing season of corn were selected: emergence, jointing, flowering, milk maturity and maturity, and the whole growing season was divided into four growth phases according to the estimated DVS. The first growth phase (from emergence to jointing), second growth phase (from jointing to flowering), third growth phase (from flowering to milk maturity), and fourth growth phase (from milk maturity to maturity) were abbreviated as GP1, GP2, GP3, and GP4, respectively.

### 4.3. Relationship between Features and Yield

Crop growth state features and environmental conditions (water, temperature) in the process of crop growth were screened and analysed, and the results are shown in Figure 6. In general, crop growth state features had a better correlation with yield, followed by water-related features. Moreover, the further the reproductive stage progresses with less certainty, the better the correlation between features and yields.

Specifically, as shown in Figure 7, the features related to the crop growth state showed a good correlation with the corn yield, which provided important crop information for the prediction of yield and were the basic features commonly used in the field of yield estimation. Among them, LAI-sum had the best comprehensive performance in the four growth phases, with a mean r of 0.50, and with crop growth, its correlation with yield gradually increased; LAI-sum in GP4 had the highest correlation with yield, with an r value of 0.69. In addition, among the three characteristic factors related to water, the TRA of the four growth stages showed a high correlation with yield, with an average r of 0.52, followed by SM, with an average r of 0.48. From the milk maturity to maturity stage, the leaves were fully developed, and the correlation coefficient between leaf transpiration and yield was 0.77. The introduction of TRA would be helpful for yield forecasting. Finally, the two features related to temperature, Tmean and Tmax, also showed a certain correlation with the yield. Among them, summer corn was greatly affected by Tmax. Compared with crop growth state and water-related features, temperature-related features do not directly show a high correlation with yield and can be used as auxiliary features to participate in yield forecasting.

Adding different types of crop information can improve yield forecasting. Considering that the features of the same type have a certain correlation, the optimal feature set was selected in each type to ensure that each type of feature can participate in the subsequent analysis to achieve a comprehensive assessment of yield forecasting capabilities. Finally, three features, LAI-sum, TRA-sum, and Tmax-sum, were selected to participate in the yield prediction.

### 4.4. Yield Prediction Potential of a Single Growth Phase and Importance Analysis of Different Features

We constructed the yield prediction model sequentially using the features of the four growth phases that were selected above. From the results (Table 5), with the continuous advancement of the crop growth stage, the accuracy of the yield prediction model continued to increase, and the ability to predict yield continued to improve. For a single growth stage, the characteristics of GP4 are crucial for the final yield formation. The crops in the first three growth phases grow rapidly, and there is great uncertainty in forecasting yield by only using the crop information of a single growth stage. The model was constructed using the characteristic factors of GP4, with the highest accuracy, R^2^ of 0.62, RMSE of 498.99 kg/ha, and MRE of 7.44%. From this point of view, the importance of the individual growth stages of GP1, 2, 3, and 4 for yield forecasting is gradually increasing. There are many uncertainties in yield estimation based on a single growth stage, the accuracy needs to be improved, and more crop information needs to be introduced to participate in yield prediction.

The analysis of the feature importance result (Figure 8) found that in the GP1 and GP2 stages, the crops were in the vegetative growth stage, LAI was unstable (which had a greater impact on the formation of yield), and LAI was the most important feature affecting the estimation of yield. In the GP3 and GP4 stages, the leaves of the crops are fully developed, gradually decline, and enter the reproductive growth stage. The main life activities are respiration and transpiration. The TRA becomes the most important feature, especially in the last growth phase, and the importance coefficient reaches 0.69.

### 4.5. Yield Prediction Potential Analysis of Multiple Growth Phase Combinations

To explore the ability of yield forecasting at different growth stages, this study designed two growth stage combinations, three growth stage combinations and whole growth stage combinations, gradually increased features, and built yield prediction models using GRU and RF models. The results are shown in Figure 9. From the performance of the RF model (Figure 9a–c)) and the GRU model (Figure 9d–f)), the GRU model outperformed the RF model at each forecasting event. With the gradual increase in growth stages and characteristics, the accuracy of yield forecasting increased, and the yield prediction ability continuously improved.

Compared with the RF method, the GRU model can better capture the dependencies in the time series feature data and improve the accuracy of yield prediction. Under the same input conditions, the GRU model has a higher R^2^, smaller RMSE and MRE, and better yield estimation performance than the RF model. The performance was particularly obvious in GP12. With the gradual addition of crop growth state, water and temperature characteristics, the RF Model R^2^ was stable at approximately 0.2, while the GRU model could extract more rules with the addition of features, and the R^2^ reached 0.53; the RMSE was 554.84 kg/ha, and the MRE was 8.27%. When the number of growth stages and characteristics increased, the advantages of the GRU model were more significant. When the GP1234 combination of LAI + TRA + Tmax was used as the model input, the R^2^ of the GRU model was 0.21 higher than that of the RF model, the RMSE was reduced by 279.07 kg/ha, and the MRE was reduced by 4.16%.

From GP12 to GP1234, the continuous addition of growth stages increases the crop information and reduces the uncertainty of the yield formation process. When using the single crop growth feature LAI for yield prediction, the R^2^ of the GRU model for yield forecasting increased from 0.23 to 0.64. The MRE decreased from 10.54% to 7.25%, the R^2^ of the RF model for yield forecasting increased from 0.20 to 0.62, and the MRE decreased from 10.54% to 7.25%. In general, the three-growth-stage combination performed better than the yield forecast model of the two-growth-stage combination, and the model for the whole growth stage performed the best. However, it is worth noting that the performance of the GP34 model is better than that of the GP123 model under some combinations of characteristics, which indicates that the participation of key growth stages in yield prediction can counteract the advantage of the number of growth phases.

Figure 10 shows that when the water and temperature features were gradually added, the yield estimation accuracy under each combination of growth stages was improved. When the LAI data of the whole growth period were input, the accuracy of the GRU model reached an R^2^ of 0.64, RMSE of 486.73 kg/ha, and MRE of 7.25%. When Tmax and TRA were added, the R^2^ of the model increased to 0.88 and 0.94, respectively, and the MRE was reduced to 4.11% and 3.06%, respectively. The water features were better than the temperature features, which contributed to the accurate prediction of yield. When the characteristics of the crop growth state, water and temperature were all involved in the yield estimation, the forecast accuracy of the yield was significantly improved, R^2^ reached 0.98, the RMSE was 102.65 kg/ha, and the MRE was 1.53%, which was also the best among all combinations.

As shown in Table 6, the optimal model used three feature combinations of crop growth state, water and temperature characteristics as the model input. Before the corn flowering period, the optimal yield prediction accuracy R^2^ reached 0.53, the RMSE was 554.84 kg/ha, and the MRE was 8.27%. Before corn milk maturity, the optimal yield prediction accuracy R^2^ reached 0.89, the RMSE was 268.76 kg/ha, and the MRE was 4.01%; this was also the best model to achieve advanced prediction of corn yield in practical applications. Using the characteristics of the whole growth period to estimate the yield, the R^2^ reached 0.98, the RMSE was 102.65 kg/ha, and the MRE was 1.53%.

## 5. Discussion

This study exploited the merits of a process-based model (WOFOST) and empirical statistical model (RF or GRU), and developed corn yield prediction models under various input combinations. These models were not only supported by agronomic mechanisms from climate, crop, soil, and management information, but also had a strong ability to capture the relationship between yield factors of each growth stage and yield, which facilitates full exploitation of the potential of summer corn yield prediction at specific growth phases.

Compared with previous studies [5,48,49], we comprehensively evaluated the performance of yield prediction at different growth phases by combining phenological information, environmental conditions and crop growth state features, instead of only using some vegetation indices of growth stages for yield prediction. Our results show that the addition of more types of features can describe the growth of crops more comprehensively, which is conducive to yield forecasting. Furthermore, the correlation of features with yield increased as the growth phase progressed, which is consistent with previous studies [50].

Advanced data-driven methods (GRU) were used in this study, the modelling process was fully trained and learned, and the yield prediction accuracy was improved. The results show that the GRU model can capture the cumulative effects of various types of features, and the model achieved higher performance in yield prediction. Multiple growth phase combinations can provide rich crop growth information, thereby alleviating the bias of single-growth-phase yield prediction. Yield prediction accuracy can be improved by combining features from multiple crop growth phases, which is consistent with previous studies [51,52].

Our study was based on a dataset that was simulated by the WOFOST model. The WOFOST model was used to dynamically simulate yield factors and yield for the entire growing season of corn in various scenarios, which provided sufficient learning information for the yield prediction model in the training process. Ensuring the rationality of the simulated dataset is critical. On the one hand, the rationality of the simulation results can be explained according to the comparative analysis of the simulated yield and the measured yield in Section 4.1. However, regarding model localization, this paper referred to calibration parameters from previous studies. In the future, actual yields can be used for calibration, thereby improving the accuracy and reliability of simulation results. On the other hand, the low simulation error of DVS at flowering and maturity reflected the correctness of the simulated dataset to some extent. Additionally, we used DVS to refine the growth phase according to the accumulated temperature threshold calculated from empirical calendar days and historical meteorological data, and the results show that the refinement of DVS is reasonable. If the actual phenological date can be obtained, the growth phases will be delineated more accurately and in detail, which could result in more than four growth phases.

In addition, we evaluated the level of yield predictions that can be achieved through a combination of yield factors at specific growth phases according to the accuracy of the prediction models. However, this study did not use measured yield data to evaluate the application of these yield prediction models due to time constraints. With the continuous development of remote sensing technology, the current multisource remote sensing data can meet the requirements of long time series yield factor data, which can be used for model application in the future.

## 6. Conclusions

In this study, we combined a crop growth simulation model (WOFOST) with deep learning methods to dynamically forecast corn yield at various growth phases, and analysed the potential of different growth phases and features to forecast corn yield.

According to the comprehensive performance of the three types of features in the four growth phases, growth state features had a better correlation with yield, followed by water-related features, and temperature-related features. Specifically, LAI-sum, TRA-sum and Tmax-sum were the optimal variables of each of the three types of yield factors. Moreover, the further the reproductive stage progressed, the better the correlation between yield factors and yields, which provided more useful crop information for the prediction of yield.

For a single growth stage, the yield prediction ability of GP1, 2, 3, and 4 gradually improves because the uncertainty in yield formation gradually diminishes. In the vegetative growth stage (GP1 and GP2) and reproductive growth stage (GP3 and GP4) of corn, the most important yield predictors are LAI and TRA, respectively.

The combination of multiple growth phases and features could provide more crop information for yield prediction. We found that the forecast accuracy improved with the increase in growth phases, and characteristics were fed into the yield prediction model. In addition, the GRU model can better capture the dependencies in the time series feature data, which outperformed the RF model at each yield forecasting event. We determined the performance of the optimal yield prediction model in flowering (R^2^ = 0.53, RMSE = 554.84 kg/ha, MRE = 8.27%), milk maturity (R^2^ = 0.89, RMSE = 268.76 kg/ha, MRE = 4.01%), and maturity (R^2^ = 0.98, RMSE = 102.65 kg/ha, MRE = 1.53%).

In this study, we explored the ability to dynamically predict yield in the continuous growth phases by a hybrid approach using a process-based model and deep learning. The approach solved the problem of sample scarcity and strengthened the mechanism of agronomy in yield forecasting, which provided new ideas for yield prediction and had the potential to be extended to other crops. Additionally, we analysed the yield prediction ability at different growth phases, combining phenological information, environmental conditions and crop growth state features, which fully illuminated the complex relationship between different types of features and yield at various growth phases. This study has provided theoretical guidance for realizing early high-precision yield prediction and agricultural decision making.

## Figures and Tables

**Figure 1 plants-12-00446-f001:**
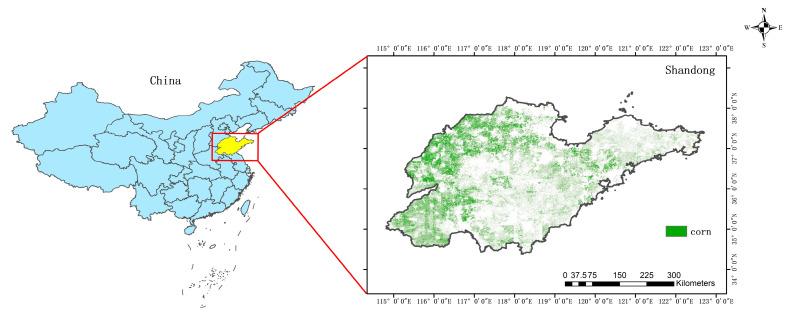
Study area. (Green represents the corn region).

**Figure 2 plants-12-00446-f002:**
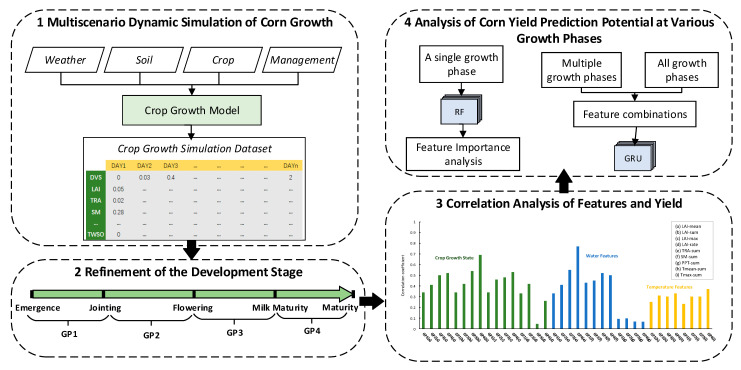
Research route of technology. RF represents Random Forest model. GRU represents the Gated Recurrent Unit model. GP represents growth phase. GP1 is from emergence to jointing, GP2 is from jointing to flowering, GP3 is from flowering to milk maturity, GP4 is from milk maturity to maturity.

**Figure 3 plants-12-00446-f003:**
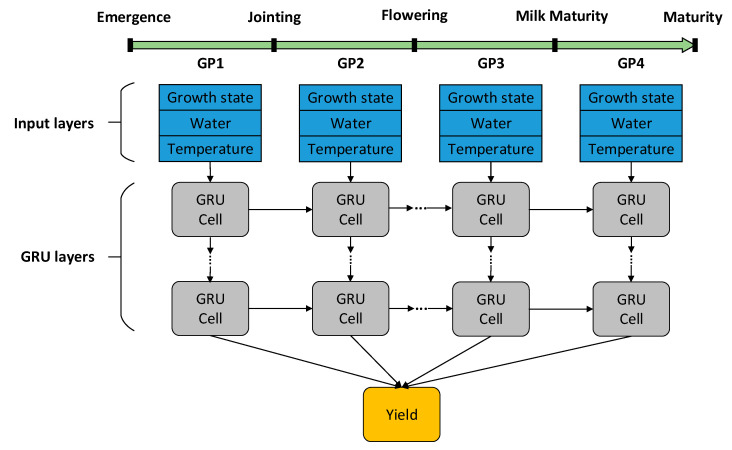
The architecture of the GRU yield prediction model. GRU represents The Gated Recurrent Unit model. GP represents growth phase. GP1: from emergence to jointing, GP2: from jointing to flowering, GP3: from flowering to milk maturity, GP4: from milk maturity to maturity.

**Figure 4 plants-12-00446-f004:**
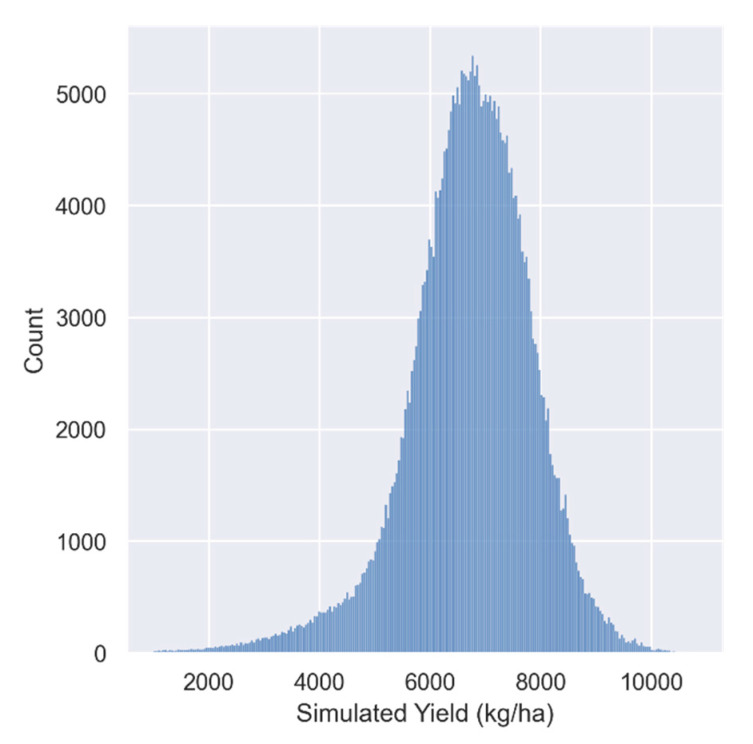
Histogram of Simulated Yields.

**Figure 5 plants-12-00446-f005:**
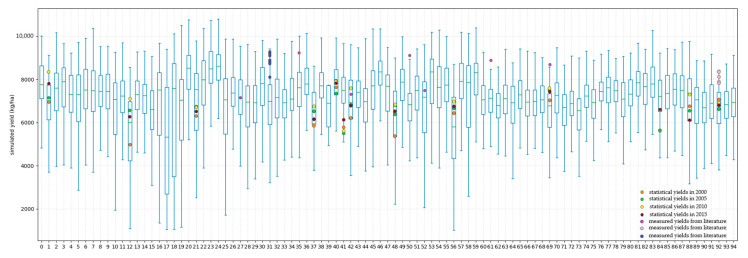
Box Plot of Simulated Yields. There are a total of 95 observation stations in the study area, and each observation station has simulated the corn yield under various growth scenarios. The measured and statistical yields of some observation stations are marked in the figure. The measured yield data is from relevant references [22,46,47].

**Figure 6 plants-12-00446-f006:**
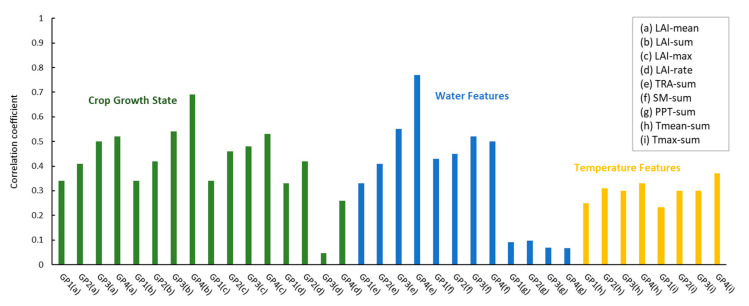
Correlation Analysis Results of Each Feature in Four Growth Phases. LAI-mean is the mean value of LAI, LAI-sum is the cumulative value of LAI, LAI-max is the maximum value of LAI, LAI-rate is the growth rate of LAI, TRA-sum is the cumulative value of TRA, SM-sum is the cumulative value of SM, PPT-sum is the cumulative value of PPT, Tmean-sum is the cumulative value of Tmean, and Tmax-sum is the cumulative value of Tmax.

**Figure 7 plants-12-00446-f007:**
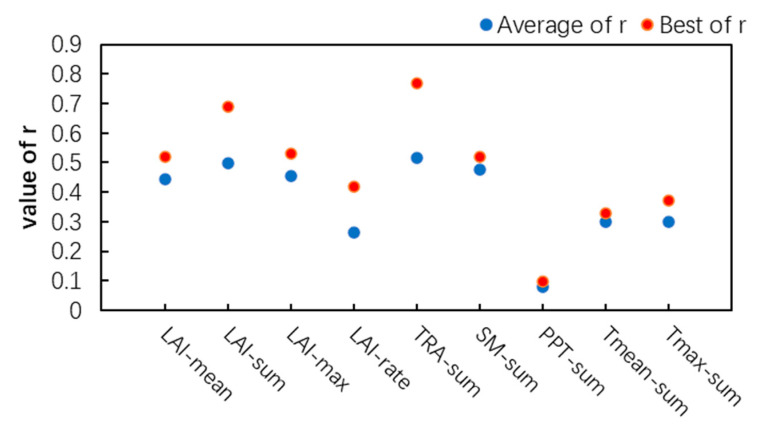
Comprehensive Performance of Each Feature.

**Figure 8 plants-12-00446-f008:**
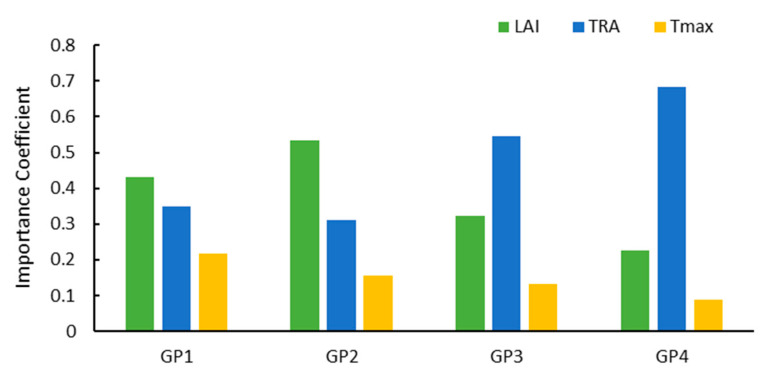
Results of Feature Importance Analysis.

**Figure 9 plants-12-00446-f009:**
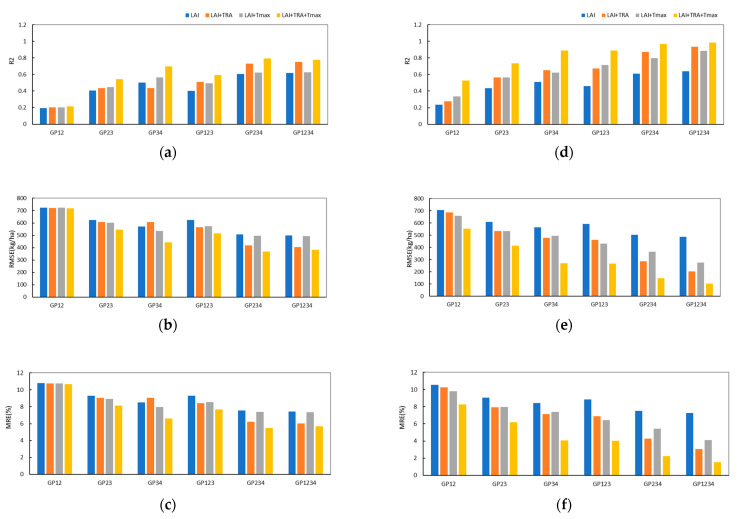
Performance of RF and GRU models at each predicted event. (**a**) The R^2^ of the RF model in every forecast event; (**b**) The RMSE of the RF model in every forecast event; (**c**) The MRE of the RF model in every forecast event; (**d**) The R^2^ of the GRU model in every forecast event; (**e**) The RMSE of the GRU model in every forecast event; (**f**) The MRE of the GRU model in every forecast event.

**Figure 10 plants-12-00446-f010:**
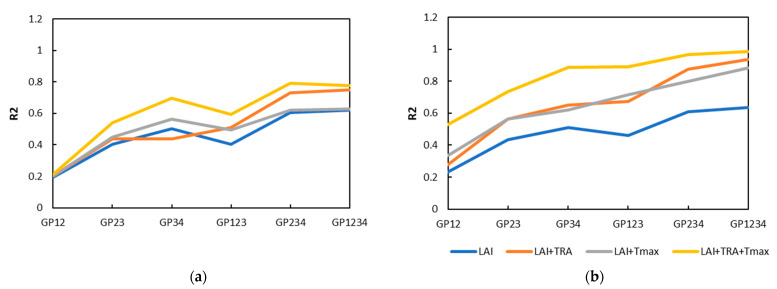
R^2^ of yield forecast models under different input variable combinations. (**a**) R^2^ of RF models; (**b**) R^2^ of GRU models.

**Table 1 plants-12-00446-t001:** Some crop parameters.

Crop Parameters	Meaning	Units	Values
TSUM1	Temperature sum from emergence to anthesis	°C	800–1000
TSUM2	Temperature sum from anthesis to maturity	°C	750–950
SLATB1	Specific leaf area (DVS = 0)	kg/ha	0.0026–0.0035
SPAN	Life span of leaves growing at 35 °C	d	35–45
TBASE	Lower threshold temperature for aging of leaves	°C	8–10

**Table 2 plants-12-00446-t002:** Main soil parameters in WOFOST.

Loam Type	SMW (cm^3^/cm^3^)	SM0 (cm^3^/cm^3^)	SMFCF (cm^3^/cm^3^)
Sandy loam	0.06	0.35	0.28
Light loam	0.09	0.34	0.28
Medium loam	0.11	0.34	0.28

**Table 3 plants-12-00446-t003:** The different combinations of growth phases and features.

Growth Phases	Features
GP1 and GP2	Growth state
	Growth state and water
	Growth state and temperature
	Growth state, water and temperature
GP2 and GP3	Growth state
	Growth state and water
	Growth state and temperature
	Growth state, water and temperature
GP3 and GP4	Growth state
	Growth state and water
	Growth state and temperature
	Growth state, water and temperature
GP1, GP2 and GP3	Growth state
	Growth state and water
	Growth state and temperature
	Growth state, water and temperature
GP2, GP3 and GP4	Growth state
	Growth state and water
	Growth state and temperature
	Growth state, water and temperature
GP1, GP2, GP3 and GP4	Growth state
	Growth state and water
	Growth state and temperature
	Growth state, water and temperature

GPs represent different growth phases. GP1 is from emergence to jointing, GP2 is from jointing to flowering, GP3 is from flowering to milk maturity, GP4 is from milk maturity to maturity.

**Table 4 plants-12-00446-t004:** Refinement Results of DVS Based on Effective Accumulated Temperature Threshold.

Phenological Period	Emergence	Jointing	Flowering	Milk Maturity	Maturity
**Calendar day**	In late June	In mid-July	In early August	In late August	In mid-September
**Estimated DVS**	0	0.45–0.47	0.95–0.99	1.47–1.54	1.93–1.99

A time point in the process of crop growth is called the phenological period, and in this study, we used 5 phenological periods to obtain 4 growth phases.

**Table 5 plants-12-00446-t005:** Performances of the yield forecast models of a single growth phase.

Growth Phase	R^2^	RMSE (kg/ha)	MRE (%)
GP1	0.003	806.91	12.03
GP2	0.18	733.50	10.93
GP3	0.19	728.49	10.86
GP4	0.69	498.99	7.44

GP1 is from emergence to jointing, GP2 is from jointing to flowering, GP3 is from flowering to milk maturity, GP4 is from milk maturity to maturity.

**Table 6 plants-12-00446-t006:** Optimal Yield Prediction Model for Combinations of Different Growth Periods.

	The Best GRU Model			The Best RF Model		
	R^2^	RMSE (kg/ha)	MRE (%)	R^2^	RMSE (kg/ha)	MRE (%)
GP12	0.53	554.84	8.27	0.21	716.49	10.68
GP23	0.74	416.31	6.21	0.54	546.89	8.15
GP34	0.89	271.40	4.05	0.70	444.36	6.62
GP123	0.89	268.76	4.01	0.59	515.15	7.68
GP234	0.97	149.09	2.22	0.79	368.27	5.49
GP1234	0.98	102.65	1.53	0.78	381.72	5.69

The input feature of the optimal model was the combination of LAI, TRA and Tmax. GP1: from emergence to jointing, GP2: from jointing to flowering, GP3: from flowering to milk maturity, GP4: from milk maturity to maturity. GP12 represents that GP1 and GP2 are combined, and GP23, GP34, GP123, GP234, GP1234 has similar meaning, respectively.

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
