# Peer review of "Analysis of Corn Yield Prediction Potential at Various Growth Phases Using a Process-Based Model and Deep Learning"

_plants, 2023, doi:10.3390/plants12030446_

Round 1

Reviewer 1 Report

This paper used a hybrind modelig approach to predict corn yields.

In abstract, please describe in a sentence why r2 is lower at flowering stage, but it increases as corn is matured.

please, for numerical results, use only two decimal point 

for example, R2=0.529--> 0.53, RMSE=554.835 --> 554.84

in introduction, line 35 should 'state' to stage?

in line 46-50, some models such as CERES and ALAMANC can simulate yield increases by simulating LAI or root growth by daily or monthly. Author should clear the word. does author mean the maturity stage, not talk about whole growth stage? Author should change growth stage to maturity stage or growth stages after flowering because there is only few models simualtes to yield increases after flowering. 

In method. 

the simulated results were not calibrated or validated by real corn yields. I think that this is the biggest limitation of the modeling approach proposed by author. 

In result section, author should convince readers that the simulated corn yields were reasonable by comparing the measured yields that can be obtained from different references or some public open access data. please find some references that support the simulated yields by process based model. 

Author Response

Dear Reviewer,

Thank you very much for your time involved in reviewing the manuscript and your comments. We have revised the manuscript according to your suggestions. The response to you is in the attachment.

Thanks again for your comments.

Yours sincerely,

Yiting Ren

Reviewer 2 Report

In general, the article entitled “Analysis of corn yield prediction potential at various growth phases using a process-based model and deep learning seems to be adequate for the journal.

I believe that the article can be mention that this is a tool that can help farmers, administrations… to take decisions. This could be part of a decision-making tool and could have a great value. Moreover, this also can be the basement for the development of this type of tool for other crops.

I think that the methodology employed is adequate and, although it is not explained in depth, authors have done a good resume. It is a complex paper in some way, because the text in results section is plenty of numerical data. But it is ok. I agree with the discussion and conclusions.

Despite these general comments, please check the following:

As a minor comment, check if you can avoid to cut the words at the end of a sentence in such way that it is difficult to read, for instance in line 68 “learn- ing” would be better in this way “Lear-ning”. The same for other lines like 175, 185,…

About the references, why are most of the references and authors from the same regional area? I am sure that the issue of the paper is of world importance and surely you have some works related to this article done by many researchers from other areas of the world. Those works would be considered.
Check them. For instance:
Hanway J J. How a corn plant develops[J]. 1966
47. Leo Breiman. Random Forests[J]. Machine Learning,2001,45(1)
Surely it is correct but, can authors provide the link to the reference? Can you complete the information required by the journal style for the references?
Thanks for improving the article. I hope you can introduce some minor changes

Author Response

(The authors gave the same response as above.)
